# Polar Infection of Echovirus-30 Causes Differential Barrier Affection and Gene Regulation at the Blood–Cerebrospinal Fluid Barrier

**DOI:** 10.3390/ijms21176268

**Published:** 2020-08-29

**Authors:** Marie Wiatr, Ricardo Figueiredo, Carolin Stump-Guthier, Peter Winter, Hiroshi Ishikawa, Ortwin Adams, Christian Schwerk, Horst Schroten, Henriette Rudolph, Tobias Tenenbaum

**Affiliations:** 1Pediatric Infectious Diseases, University Children’s Hospital Mannheim, Medical Faculty Mannheim, Heidelberg University, 68167 Mannheim, Germany; marie.wiatr@medma.uni-heidelberg.de or carolin.stump-guthier@medma.uni-heidelberg.de (C.S.-G.); christian.schwerk@medma.uni-heidelberg.de (C.S.); horst.schroten@umm.de (H.S.); henriette.rudolph@medma.uni-heidelberg.de or; 2GenXpro GmbH, 60438 Frankfurt am Main, Germany; rfigueiredo@genxpro.de (R.F.); pwinter@genxpro.de (P.W.); 3Johann Wolfgang Goethe University Frankfurt, 60438 Frankfurt Am Main, Germany; 4Department of Clinical Regenerative Medicine, Department of Neurosurgery, Faculty of Medicine, University of Tsukuba, Tennodai, Tsukuba, Ibaraki 305-0005, Japan; ishi-hiro.crm@md.tsukuba.ac.jp; 5Institute for Virology, Heinrich Heine University, 40225 Düsseldorf, Germany; ortwin.adams@uni-duesseldorf.de

**Keywords:** blood–cerebrospinal fluid barrier, Echovirus-30, meningitis, polarity, MACE RNA sequencing

## Abstract

Echovirus-30 (E-30) is responsible for the extensive global outbreaks of meningitis in children. To gain access to the central nervous system, E-30 first has to cross the epithelial blood–cerebrospinal fluid barrier. Several meningitis causing bacteria preferentially infect human choroid plexus papilloma (HIBCPP) cells in a polar fashion from the basolateral cell side. Here, we investigated the polar infection of HIBCPP cells with E-30. Both apical and basolateral infections caused a significant decrease in the transepithelial electrical resistance of HIBCPP cells. However, to reach the same impact on the barrier properties, the multiplicity of infection of the apical side had to be higher than that of the basolateral infection. Furthermore, the number of infected cells at respective time-points after basolateral infection was significantly higher compared to apical infection. Cytotoxic effects of E-30 on HIBCPP cells during basolateral infection were observed following prolonged infection and appeared more drastically compared to the apical infection. Gene expression profiles determined by massive analysis of cDNA ends revealed distinct regulation of specific genes depending on the side of HIBCPP cells’ infection. Altogether, our data highlights the polar effects of E-30 infection in a human *in vitro* model of the blood–cerebrospinal fluid barrier leading to central nervous system inflammation.

## 1. Introduction

The entry of various pathogens into the central nervous system (CNS) can lead to meningitis and encephalitis in humans [1,2]. Besides human pathogenic bacteria, such as *Neisseria meningitidis (N. meningitidis)* and *Streptococcus pneumoniae* [3,4], viruses, such as enterovirus, can lead to neuroinflammation, especially in countries with a high pneumococcal and meningococcal vaccine uptake [5,6,7]. Among the enteroviruses, Echovirus-30 (E-30) is a nonpolio enterovirus (NPEV) species B, responsible for extensive global outbreaks of meningitis in children [8,9,10], and which can result in mild symptoms, but might also rapidly evolve towards a lethal outcome [11,12]. Several clinical studies revealed that enterovirus infection of the CNS results in increased levels of inflammatory chemokines within the cerebrospinal fluid (CSF), including *INF-γ*, *CXCL12*, *CXCL10*, *IL-6*, *IL-1* and *TNF-α*, which increase the influx of neutrophils and T cells into the brain parenchyma [13,14,15]. However, the underlying mechanisms used by these viruses to enter the brain and induce CNS inflammation are not fully understood yet. 

To gain access to the CNS, pathogens need to cross the main barriers, such as the blood–brain barrier (BBB) or the blood–CSF barrier (BCSFB), where they can cause inflammation. It was previously shown in human and porcine *in vitro* models of the BCSFB that infection with specific bacterial pathogens can alter the properties of the BCSFB and lead to invasion of pathogens into the brain [16,17,18]. Remarkably, *Streptococcus suis* and *N. meningitidis* were found to invade respectively a porcine and human *in vitro* model in a polar manner [19,20], a phenomenon that has not been investigated in viral CNS infection pathogenesis yet. Previous publications have shown that the polarity of epithelial cells may also be critical for enterovirus infection, such as for Coxsackievirus B3 and Echovirus 7 [21,22]. 

In this study, we have investigated the polar infection of human choroid plexus papilloma (HIBCPP) cells with E-30 by focusing on the impact on BCSFB properties and gene regulation in this *in vitro* model. In previous experimental work, we showed that HIBCPP could efficiently be infected with E-30 from the apical as well as from the basolateral side [23]. In the current work, we first demonstrated that basolateral infection had a greater impact on the transepithelial electrical resistance of HIBCPP cells compared to the apical infection. Furthermore, HIBCPP cell viability following polar E-30 infection was decreased during basolateral compared to apical infection, and by means of immunofluorescence analysis, the number of infected HIBCPP cells was found to be lower in apical compared to basolateral infection. Lastly, a massive analysis of cDNA ends (MACE) on RNA extracted from HIBCPP cells following basolateral and apical infection with E-30, revealed differentially expressed genes depending on the side of infection.

## 2. Results

### 2.1. Higher Impact on Barrier Integrity Following Basolateral Infection of HIBCPP Cells with E-30 Compared to Apical Infection

The impact of apical or basolateral infection of HIBCPP cells (Figure 1) with an increasing MOI (multiplicity of infection) of E-30 at different time-points was investigated. The transepithelial electrical resistance (TEER) was measured at 8, 16 and 24 h postinfection from the basolateral or apical side and this for low to high MOI of E-30, such as 0.7, 1.4, 2.1, 9.1, 14 and 20. There was no significant decrease in the TEER following 8 h of infection from the basolateral or the apical side of HIBCPP cells compared to the uninfected control condition, even with a high MOI of 20 (Figure 2A,B). A significant drop of the TEER was observed at 16 h postinfection from the basolateral side at MOI 2.1, 9.1, 14 and 20, compared to the TEER at 0 h (*p* = 0.013, *p* = 0.0081, *p* < 0.0001 and *p* < 0.0001, respectively; Figure 2C), but not after apical infection (Figure 2D). Finally, after 24 h of infection, we observed a significant decrease in TEER for all the conditions following the basolateral infection of HIBCPP with E-30 (Figure 2E). The TEER drop after 24 h of basolateral infection with E-30 on HIBCPP cells was more significant with higher MOI (i.e., 9.1, 14 and 20) compared to lower MOI (i.e., 0.7, 1.4 and 2.1; *p* = 0.0006, *p* < 0.0001, *p* < 0.0001 and *p* = 0.027, *p* = 0.0058, *p* = 0.018, respectively; Figure 2E). Nevertheless, when HIBCPP cells were apically infected with E-30 for 24 h, only the highest MOI (i.e., 14 and 20) showed a significant TEER decrease compared to the uninfected control condition (*p* = 0.0044, *p* = 0.0006, respectively; Figure 2F). In conclusion, a significant decrease in TEER was observed following basolateral infection starting from an already lower MOI and shorter incubation period compared to the apical infection.

### 2.2. E-30 Has a Greater Cytotoxic Effect after Basolateral Compared to Apical Infection

Earlier publications demonstrated that prolonged E-30 infection from the basolateral side of HIBCPP cells was leading to cell death [24,25]. Nevertheless, the infection of HIBCPP cells with E-30 from the apical side has never been investigated in this context. Therefore, the cytotoxic effects of basolateral versus apical E-30 infection of HIBCPP cells were evaluated in a dose and time-dependent manner via Live/Dead viability assay. In the previous section, no TEER decrease at 8 h postE-30 infection was observed (Figure 2A,B). Under this condition, the number of dead cells at 8 h postinfection with E-30 was very low, and this regardless of the MOI used and of the side of infection (Figure 3). Further, at 16 h postinfection, a slight increase in the number of dead cells was observed for MOI 2.1 and MOI 9.1, preferentially after basolateral infection (Figure 3A). However, when HIBCPP cells were challenged for 24 h with E-30 from the basolateral side (Figure 3A), a more pronounced number of dead cells was detected compared to challenge from the apical side (Figure 3B), especially for infections at a higher MOI. Specifically, HIBCPP cells infected from the basolateral side for 24 h at MOI 20 (Figure 3A) showed a drastic number of dead cells compared to HIBCPP cells from the apical side for the same condition, where almost no cytotoxicity could be observed (Figure 3B). Taken together, Live/Dead viability assays showed that HIBCPP cells were more susceptible to cytotoxicity after basolateral E-30 infection compared to apical infection.

### 2.3. Basolateral Infection of HIBCPP Cells with E-30 Resulted in a Higher Number of Infected Cells Compared to Apical Infection

Analysis of TEER values and cell viability indicated that basolateral infection had a more pronounced impact on HIBCPP cell integrity than apical infection. In the next set of experiments, the number of HIBCPP cells that were infected with E-30 following basolateral or apical infection was evaluated via immunofluorescence imaging at time-points 16 and 24 h postinfection. Immunofluorescence analysis showed a higher number of infected HIBCPP cells when cells were infected basolaterally with MOI 0.7 for 24 h compared to apical infection (Figure 4A,B). Furthermore, immunofluorescence quantification at 16 h postinfection demonstrated a significantly higher number of infected HIBCPP cells after basolateral infection with MOI 0.7 and MOI 2.1 compared to apical infection (*p* = 0.0029, *p* = 0.0015, respectively; Figure 4C). Moreover, at the 24 h time-point, the number of HIBCPP cells infected with E-30 from the basolateral side was significantly higher compared to apical challenge for both MOI tested (MOI 0.7 *p* = 0.0216 and MOI 20 *p* = 0.007; Figure 4D). Importantly, after 24 h, approximately the same number of E-30 infected HIBCPP cells were observed after basolateral infection with a MOI of 0.7 as with a MOI of 20 after apical infection (27.08 ± 9.25% and 26.29 ± 6.47%, respectively; Figure 4D). Taken together, these results demonstrated that HIBCPP cells are infected with E-30 more efficiently from the basolateral side when compared to apical infection.

### 2.4. Apical and Basolateral Infection of HIBCPP Cells with E-30 Leads to Specific Differentially Expressed Genes

Next, the impact of apical and basolateral E-30 infection of HIBCPP cells was evaluated at the RNA level. Gene expression profiling analysis of either apically or basolaterally infected HIBCPP cells was performed in order to reveal specific impacts due to cell polarity. Results in the previous section showed that apical infection of HIBCPP cells at a MOI of 20 lead to approximately the same number of infected cells as infection with a MOI of 0.7 from the basolateral side (Figure 4). Therefore, the impact of apical versus basolateral infection with MOI 20 and MOI 0.7, respectively, was evaluated by analyzing the gene expression of HIBCPP cells following infection with these MOI. After polar infection of HIBCPP cells, considerable changes in the transcriptomic profiles were observed (Figure 5A). Consequently, a large number of biological pathways were affected (Figure 5B). At first, we compared the inverted to the standard cell culture system under uninfected conditions to exclude differentially expressed genes that did not solely depend on viral infection. Comparison between both culture systems revealed a low number of differentially expressed genes in HIBCPP cells, and thus only small differences exist on the transcriptional level between the two culture models (Figure 5 and Figure 6A). 

Further analysis of the sequencing data revealed 195 differentially expressed genes when basolaterally infected HIBCPP cells were compared to uninfected control cells (Figure 5 and Figure 6B, Appendix A). We noted a strong upregulation of genes encoding for *CXCL2*, *CXCL3*, *CXCL10*, *IFNβ-1*, *IFNλ-1*, *-2*, *-3*, endothelin 1 (*EDN1*) and *IFIT-1*, *-2*, *-3* following E-30 infection (Figure 7A, Appendix A). The chemokines *CXCL2*, *CXCL3* and *CXCL10* are part of “the immune response” pathway, which is highly upregulated following basolateral infection compared to control, as shown in the gene set enrichment analysis (GSEA; Figure 8). Additionally, the interferon genes, such as *IFN-λ-1*, *-2* and *-3*, which confer a strong “defense against viral infection” were upregulated in HIBCPP cells after basolateral infection (Figure 8). Interestingly, the gene encoding for endothelin 1 *EDN1*, a vasoconstrictor, which is usually secreted by endothelial cells, was upregulated by HIBCPP cells under both infectious conditions; *EDN2* was differentially expressed solely following basolateral infection (Figure 7A and Figure 8, Appendix A). We further observed a downregulation of the “glycogen process” pathway due to the differential expression of genes such as hexokinase (*HK2*; Figure 7A and Figure 8). Additionally, genes implicated in cation channel activity were differentially expressed after basolateral infection leading to downregulation of the “voltage-gated cation channel activity” pathway (Figure 8). Furthermore, genes involved in mitochondrial activity, such as *MIGA-1*, were differentially expressed after basolateral infection of HIBCPP cells compared to control conditions (Figure 7A and Figure 8). Finally, the downregulation of the “neuronal cell body” pathway was due to differential expression of genes such as myosin (*MYH10*) at the choroid plexus (Figure 8, Appendix A). 

Analysis of apically infected HIBCPP cells resulted in 226 differentially expressed genes (Figure 6C; Appendix A). We found upregulated genes that were also upregulated following basolateral infection, such as *CXCL2*, *CXCL3*, *CXCL10*, *IFNβ-1*, *IFNλ-1*, *-2*, *-3*, endothelin 1 (*EDN1*) and *IFIT-1*, -*2* and *-3* (Figure 7B, Appendix A). Interestingly, we noted the additional upregulation of *CXCL1* and *CXCL8*, which are critical for the recruitment of neutrophils at the choroid plexus (pathway called “positive regulation of neutrophil migration”; Figure 7B and Figure 9). *CCR1*, a receptor involved in chemotaxis of immune cells at the site of inflammation, was upregulated after apical infection (Figure 7B, Appendix A). IL-6, a critical inflammatory cytokine, was also upregulated (Figure 7B, Appendix A). Altogether, these genes are part of the cytokine-mediated signaling pathway, viral defense response, response to stress and regulation of programmed cell death, highlighting the diverse molecular response of the HIBCPP cells after E-30 infection (Figure 9). 

Finally, the direct comparison of basolateral infection at MOI 0.7 versus apical infection at MOI 20 of HIBCPP cells was analyzed (Figure 6D, Appendix A). Expression of genes implicated in angiogenesis and cytoskeleton organization was increased following basolateral infection compared to apical infection (Figure 10). Among these genes, we found actin like 10 (*ACTL10*), and cell division control protein 42 homolog (*Cdc42 rho GTPase*; Figure 7C, Appendix A). Additionally, in basolateral infection, we noticed a relative downregulation of many important genes, when compared to apical infection such as *CCR1*, *CXCL10*, integrin α5 (*ITGα5*) and *IFNλ-1, -2* and *-3*. These genes have a critical role in pathways such as cell adhesion, regulation of the cellular metabolic process and intracellular signal transduction (Figure 10).

At last, we confirmed the regulation of selected genes via quantitative PCR. We verified a strong upregulation of *CXCL2*, *CXCL3*, *IFNλ-1*, *-2* and *IFIT-2* genes following a basolateral infection of HIBCPP with E-30 at MOI 0.7 compared to uninfected conditions (Figure 11A, Appendix A). Furthermore, we verified the upregulation of *CXCL8, CXCL2, CXCL3, IFNλ-1, -2* and *IFIT-2* genes following an apical infection of HIBCPP cells with E-30 at MOI 20 compared to uninfected condition (Figure 11B, Appendix A). Under both conditions, we could not clearly show the downregulation of *ITGα5* and *EDN2* in HIBCPP cells infected with E-30 (Figure 11). Furthermore, the upregulation of *IL-6* following apical or basolateral E-30 infection was observed with the q-PCR but was not significant (Figure 11). Taken together, these results confirm differential gene regulation of HIBCPP cells depending on basolateral or apical infection indicating the involvement of distinct pathways.

## 3. Discussion

### 3.1. Basolateral Infection Has a Drastic Impact on the BSCFB Properties

E-30 is a highly infective enterovirus with a diverse cell tropism [27]. Enteroviruses, in general, can multiply in various cell types, including neurons, endothelial and epithelial cells [28]. Still, the specific receptors and pathways by which E-30 enters the respective cells remain partially unresolved [29]. 

Several studies showed that the pathways used by enteroviruses to enter a polarized cell or a nonpolarized cell are divers [30,31]. In this article, we infected the basolateral or apical side of HIBCPP cells with E-30 to monitor the differential influence on HIBCPP properties. Remarkably, we observed a polar infectability of HIBCPP cells as well as polar effects on the cellular and molecular host response. We could demonstrate an impact on the TEER as a measure for barrier integrity depending on the E-30 MOI and stimulation time. To reach the same impact on barrier integrity, the MOI added to the apical side needed to be more than twenty times higher, than that added to the basolateral side of the cells. During the early phases of infection, such as 8 h, the TEER of HIBCPP cells was stable following an apical and basolateral E-30 infection. Similar results have been observed in an *in vitro* model of the BBB following an early stage of infection with E-30 or other Echoviruses [32]. In human BBB models applying human brain microvascular endothelial cells (HBMEC) or human cerebral microvascular endothelial cells (hCMEC/D3), it was shown that infection with poliovirus [33], E-6, E-12 and E-30 [32] resulted in damage to the junctional connections, leading to increased paracellular permeability of the barrier. Therefore, enterovirus that have crossed the BBB and that are circulating in the CSF may apically enter the choroid plexus epithelium. Moreover, a reseeding of enterovirus that have entered via the BSCFB into the bloodstream is also feasible. 

In the current study, we reached a similar high TEER decrease with a high MOI added to the apical side after a long virus exposition (e.g., 24 h). Based on these results, we suspect that the receptors required for E-30 infection may be more abundant or accessible to the virus at the basolateral side. In a recent study, it was shown that basolateral infection of HIBCPP cells with different E-30 outbreak strains had a variable impact on barrier integrity, which went along with alterations of tight and adherens junctions. This effect was also depending on the virulence of the E-30 strains [24]. Taken together, our results indicate that E-30 can infect HIBCPP cells from both sides; however, the TEER decrease accompanied by apical infection was delayed and less effective compared to the basolateral infection. 

The limitation of our model is that we used HIBCPP cells as a BCSFB model and not primary human choroid plexus epithelial cells. However, our model is currently the only human model existing. HIBCPP may not represent all properties compared to primary cells and therefore our results should be interpreted with caution. Several rodent models of BCSFB have been established, including immortalized rat, as well as primary mouse choroid plexus epithelial cells [34,35]. Primary porcine choroid plexus epithelial cell cultures have been used to study bacterial infection of the brain and have demonstrated polar infection [19]. A novel porcine choroid plexus epithelial cell line (PCP-R) has also been established [36]. Moreover, the BCSFB model with epithelial cells from Indian origin Rhesus macaques has recently been described [37].

### 3.2. Apical Infection with Echovirus 30 Leads to Minor Cytotoxicity on HIBCPP Cells Compared to Basolateral Infection

Looking at cytotoxicity, we only observed a minor impact of apical (MOI 20) and basolateral (MOI 0.7) infection of HIBCPP cells after 24 h. However, we demonstrated that a prolonged infection, especially with a higher MOI from the basolateral side, was correlated with an enhanced cytotoxicity compared to the apical side. Interestingly, in primary endothelial cells, such as human umbilical vein endothelial cells (HUVEC), or in immortalized endothelial cells, such as human dermal microvascular endothelial cells (HDMEC), infection with different strains of Coxsackievirus B (CVB) was not leading to significant cell death [38,39]. Additionally, infection with the same virus had a different impact depending on the cell type. Infection with the CVB3 enterovirus in Hela cells was leading to necrosis, while infected Caco-2 cells were undergoing apoptosis [40]. Additionally, infection with EV-71 was leading to a cytopathic effect in human rhabdomyosarcoma cells [41].

### 3.3. HIBCPP Cells Are More Susceptible to Basolateral than Apical E-30 Infection 

Our investigations revealed that the number of infected HIBCPP cells was significantly higher following basolateral infection compared to apical infection. An explanation for this phenomenon may be a delayed entry into polarized epithelial cells due to lower presence of the receptors for the virus on the apical side. Some articles highlighted the importance of coreceptors at the membrane for an efficient virus entry inside the cells. In fact, coreceptors concentrate the viral particles and enable the fixation of the virus with the receptors, which in turn induce intracellular signaling [42,43]. Overall, most publications covering enterovirus infections have analyzed only apical infections, as for example in Caco-2 cells, in which enteroviruses enter the cells via receptors present on the apical side of the cells, without taking into consideration the infection from the basolateral side of epithelial cells [33,40,44,45]. Therefore, our inverted cell culture BCSFB model is an elegant tool to investigate polar infection. 

### 3.4. Basolateral and Apical Infection Lead to Differentially Expressed Genes and Pathways

Analysis of the MACE RNA sequencing data revealed differentially expressed genes following apical and basolateral infection compared to control conditions. Following E-30 infection and regardless of the side of infection, we noted the overexpression of genes such as *CXCL2*, *CXCL3*, *CXCL10, IFNβ-1, INFλ-1, -2, -3, IFIT-1, -2, -3* and *EDN1*, which are commonly upregulated upon infection, and, therefore, indicate important defense mechanisms by HIBCPP and other epithelial cells [46,47]. A previous study has shown that basolateral infection of HIBCPP cells with E-30 resulted in an increased expression of *CXCL2* and *CXCL3*, two chemokines that play a role during inflammation processes [23]. Here, both basolateral and apical infection led to the upregulation of *CXCL2* and *CXCL3* in HIBCPP cells. Interestingly, among the unique upregulated genes upon apical infection of HIBCPP cells, we noted *CXCL1, CXCL8* and *IL-6*. Along the same lines, other studies proved that *CXCL8, CXCL1* and *IL-6* were required to induce migration of neutrophils and other immune cells to injured brain parenchyma, through the BBB and through the choroid plexus [48,49]. Furthermore, in the CSF of patients infected with E-30, an elevated concentration of *CXCL1* and *IL-6* was quantified [50]. Taken together, these results highlight an active role of HIBCPP cells in chemokine expression for chemotaxis of immune cells, which correlates with the increased number of immune cells and neuroinflammatory mediators, such as *IL-6* and *IL-β1* at the choroid plexus in vivo [51,52]. Moreover, the expression of *CXCL8* induced the relocalization of the viral receptor CAR, as well as of ανβ3 integrins to the apical side of the epithelial cells, thus contributing to adenovirus infection [53].

Additionally, the upregulation of *IFNβ-1*, *INFλ-1*, *-2*, *-3* and *IFIT-1*, *-2*, *-3*, following apical and basolateral infection was noted. In general, type I interferons are critical for defense mechanisms of host cells, due to their antiviral activity [54]. In a murine model of encephalitis, a major role of interferon type I secretion at the BCSFB during the course of the disease has been demonstrated [50,55,56]. Additionally, the susceptibility of newborns to encephalitis was due to the low secretion of interferon type I at the choroid plexus, which led to the viral spread in the newborn brains compared to adults [56]. Additionally, interferon genes were upregulated at the BCSFB in mice *in vivo*, thereby increasing the migration of CD4^+^ T cells [57]. Interestingly, endothelin 1, a protein usually secreted by endothelial cells, was upregulated after E-30 infection in HIBCPP cells, suggesting a yet unknown role of endothelin secretion, at the choroid plexus. At the BBB level, endothelin secretion was linked to the development of multiple sclerosis by favoring the passage of monocytes across this barrier [58]. However, the primary role of endothelin 1 is to promote vasoconstriction [59]. The endothelial cells at the choroid plexus are forming fenestrated capillaries, and, therefore, allow a higher blood flow [60]. It is tempting to speculate that the BCSFB secrete high levels of endothelin 1 under E-30 infection, which may lead to reduced blood flow in the choroid plexus in order to decrease further viral dissemination in the brain parenchyma. 

Moreover, we identified several genes, encoding for proteins such as endothelin 2 that are downregulated in HIBCPP cells specifically following basolateral infection. Endothelin 2 was described as an inflammatory factor relevant to the remyelination in the CNS [61]. In a murine model, it was demonstrated that a virus causing encephalitis, such as Theiler’s virus, led to demyelination of the neurons [62]. The downregulation of genes implicated in the homeostasis of the brain in HIBCPP cells following a basolateral infection with E-30 hints at a crucial role of the choroid plexus in neuronal maintenance. 

Additionally, genes mainly involved in “glycogen process”, “cation activity” and “cytoskeleton organization” were also differentially expressed after a basolateral infection. Among these was the gene encoding for hexokinase 2 (*HK2*). In literature, inhibition of *HK2* in lung epithelial cells led to a decreased number of metapneumovirus present in these cells *in vitro* [63]. Metabolic pathways, such as those involved in glycolysis, are critical for viral replication [64]. The observed downregulation of genes involved in metabolic processes in basolateral E-30 infected HIBCPP cells suggests a process of cell exhaustion, which would eventually lead to cellular apoptosis. Nevertheless, this fact could explain the severity of E-30 basolateral infection over the apical infection. Another previous study highlighted the role of the choroid plexus in regulating iron metabolism during the inflammatory process, by secreting iron homeostasis related genes, and, therefore, restricting iron availability [51]. In addition, upon basolateral infection, we identified a decreased expression of genes critical for mitochondrial activity, such as *MIGA1*. Some publications demonstrated the exploitation of mitochondrial bioenergetics by viruses [65]. Additionally, *MIGA1* has a role in neuronal homeostasis by regulating the fusion of mitochondria [66].

A direct comparison between basolateral versus apical infection of HIBCPP cells also revealed many differentially expressed genes. Cytoskeleton-related genes such as *ACTL10* and *Cdc42* were overexpressed in basolateral infection compared to apical infection. Interestingly, *Cdc42* is mainly involved in the maintenance of epithelial cell polarity [67]. On the other hand, we noticed a relative downregulation of many important genes upon basolateral infection compared to apical, *CCR1, ITGα5* and *IFNλ-1, -2* and *-3*. As previously described, these genes are involved in cell adhesion, regulation of cellular metabolic processes, and intracellular signal transduction. Type 3 interferon genes (*IFN-λ*) have been shown to have potent antiviral activity. In the gut, secretion of type 3 interferons by epithelial cells acts as a critical regulator of the innate immune system, which controls viral infection [68,69]. Based on these results, we could stipulate that the increased expression of type 3 interferon genes might be a factor involved in the polar effect of E-30 on HIBCPP cells.

In our context, basolateral infection of HIBCPP cells leads to a lower expression of *CCR1* compared to apical infection. *CCL5*, a *CCR1* ligand, is a molecule secreted by activated T cells which leads to T cell chemotaxis and, therefore, the migration of immune cells at the site of infection [70,71]. *CCR1* was shown to be upregulated in the CNS following virus infection or certain neurological diseases such as multiple sclerosis [72].

Taken together, our MACE RNA sequencing data revealed a strong impact of apical and basolateral infection by E-30 on HIBCPP cells. Furthermore, our results demonstrated a differential impact of basolateral versus apical infection, resulting in distinct expression patterns, pointing to involvement in several different cellular pathways, such as inflammation, cytoskeleton modulation, CNS homeostasis and metabolic processes. 

## 4. Materials and Methods 

### 4.1. HIBCPP Cells Culture

HIBCPP cells are human cells isolated from a 29-year-old woman, suffering from a choroid plexus tumor in the right lateral ventricle (fourth ventricle) [73]. HIBCPP cells were cultured as previously described [16]. In brief, they were cultured in a T flask (175 cm^2^; Greiner, Frickenhausen Germany) in HIBCPP 10% medium. HIBCPP cells were concentrated at 10^6^ cells/mL, and further seeded at 80,000 HIBCPP cells on filter inserts (Appendix A). After one day, the filters in the inverted culture were flipped, transferred to a 24-well plate, and HIBCPP 10% medium was added to the upper and bottom compartment of the filter insert (Appendix A). Concerning the normal culture system, after one day, HIBCPP 10% medium was added to the upper and the bottom compartment of the filter insert. After two to three days in culture, HIBCPP cells that reached a transepithelial electrical resistance (TEER) of 50 Ω × cm^2^ were switched to HIBCPP 1% medium. Cells were ready to be used in experiments 24 h after this last medium change. The barrier integrity of the HIBCPP cell layer was evaluated via TEER measurement with a tissue voltohmmeter (Milicell© ERS-2 Epithelial Voltohmmeter, Millipore, Germany). In order to perform experiments, HIBCPP cells had to reach a high TEER of 215-775 Ω x cm^2^.

### 4.2. Reagents

HIBCPP 10% medium consisted of Dulbecco’s modified eagle medium (DMEM)/Ham’s F12 1:1 medium (Life technologies, Birchwood, UK) supplemented with 4 mM L-Glutamine, 5 μg/mL insulin, 10% fetal calf serum (FCS; Gibco, Gaithersburg, MD, USA) and 2.5% Pen/Strep (MP Biomedicals, Irvine, CA, USA), HIBCPP 1% medium consisted of DMEM/Ham’s F12 1:1 medium, supplemented with 4 mM L-Glutamine, 5 μg/mL insulin and 1% heat-inactivated FCS. Cell inserts Sarstedt, Germany; pore diameter 5.0 μm, pore density 6.0 × 10^5^ pores/cm^2^, 0.33 cm^2^) in the inverted (CytoOne© 12-well plate StarLab, Hamburg, Germany) or normal (CytoOne© 24-well plate StarLab, Belgium) were used in the culture model. PBS (Gibco, Thermo Fisher, Waltham, MA, USA). Primary antibody, monoclonal mouse light diagnostics™ anti-PAN Enterovirus (Merck, Darmstadt, Germany). Secondary antibody, Alexa Fluor anti-mouse 488, and phalloidin Alexa Fluor 660 (Molecular Probes, Eugene, OR, USA). Live/Dead viability assays (Thermo Fisher Scientific, Waltham, MA, USA). RNeasy™ Micro Kit Quick Start protocol (QIAGEN, Hilden, The Netherlands). MACE-Seq kit v. 2.0 (GenXPro GmbH, Germany). Qubit HS dsDNA assay (Thermo Fisher Scientific, Waltham, MA, USA). immobilization beads (Agencourt AMPure XP, Beckman Coulter, Brea, CA, USA).

### 4.3. Virus Preparation

E-30 strain *Bastianni* was isolated from a patient suffering from aseptic meningitis in 1958. It was obtained from the National Reference Center for Poliomyelitis and Enteroviruses (NRCPE) at the Robert Koch Institute (RKI, Germany) and propagated as published earlier [23,25] using confluent rhabdomyosarcoma cells (RD cells). Confluent cultures of RD cells were infected with 100 μL of E-30 suspension. When a cytopathic effect of 90% was reached on RD cells, the suspension containing the virus was frozen overnight at −20 °C. The suspension was centrifuged for 15 min at 4000 rpm and at 4 °C, subsequently further aliquoted and frozen at −80 °C. The virus concentrations were determined via q-PCR analysis which was performed with quantitative TaqMan-real-time PCR using a well-described two-step-RNA-PCR protocol [23,25].

### 4.4. Polar Infection of HIBCPP Cells with E-30

HIBCPP cells were grown on cell culture inserts in the inverted cell culture model. Once they reached TEER values between 215–775 Ω × cm^2^, they were infected with an increasing MOI of E-30. All experiments were performed in 1% medium. To the upper compartment of the filter, 400 μL of 1% medium containing E-30 *Bastianni* at a MOI of 0.7, 1.4, 2.1, 9.1 or 20 were added and kept throughout the experiments. For uninfected controls, only 1% medium was added to the upper compartment. A schematic representation of the experimental set-up is displayed in Figure 1. Further, the barrier integrity of the HIBCPP cell layer was evaluated via TEER measurement at 8, 16 and 24 h postinfection for all MOI.

### 4.5. Quantification of HIBCPP Cells Infected with E-30

At the end of the migration assay, HIBCPP cells were rinsed in PBS and fixed in 3.7% formaldehyde for 15 min at room temperature (RT). HIBCPP cells were further washed in PBS culture insert filters were cut out and HIBCPP cells were permeabilized with 1% Triton-X-100 PBS, at RT for 20 min. Blocking of unspecific binding sites was performed by incubating the cells in 1% PBS/BSA solution for 15 min at RT. Afterwards, incubation with the primary antibody, monoclonal mouse light diagnostics™ anti-PAN Enterovirus (Merck, Germany), was performed at a dilution of 1:250 and maintained overnight at 4 °C. Then, cells were washed in PBS and incubated with the secondary antibody, Alexa Fluor anti-mouse 488, and simultaneously with phalloidin Alexa Fluor 660 (Molecular Probes, USA; staining of actin fibers), as well as 4′-6-diamidino-2-phenylindole dihydrochloride (DAPI; staining of nuclei), diluted in 1% PBS/BSA at RT for 1 h. Once this second incubation was completed, cells were washed in PBS and mounted with antifade reagent (Life Technologies, USA). At the end of the staining protocol, the quantification of infected HIBCPP cells was performed by counting the number of infected cells present in ten fields of view (FOV) randomly chosen. Images were taken with a Zeiss Apotome together with Zen software (Carl Zeiss, Oberkochen, Germany), using a magnification of 63x/1.4 NA objective lens. 

### 4.6. Cell Viability Experiments

After infection with E-30, HIBCPP cells on filter inserts were washed with serum-free medium (SFM). Next, Live/Dead viability assays were performed following the manufacturer’s instructions. In brief, a master mix of 4 µM of Ethidium homodimer-1 (staining of dead cells in red) and 0.50 µM of calcein green (staining live cells in green) was diluted in SFM. Afterwards, 100 µL of this solution was added to the upper compartment of the filter insert part of the filter, while 500 µL of the same mix was added to the bottom compartment of the filter insert. Subsequently, HIBCPP cells were placed in the incubator for 15 min at 37 °C/5% CO_2_. At the end of the incubation time, HIBCPP cells were rinsed in SFM and further analyzed using Zeiss Apotome and Zen software (Carl Zeiss, Germany), using a 10×/1.4 NA objective lens. Six FOV of the HIBCPP cell layer were taken randomly as described under “quantification of HIBCPP cells infected with E-30”.

### 4.7. Quantitative RT-PCR

Prior to RNA isolation, HIBCPP cells were rinsed twice in PBS. RNA was isolated from HIBCPP cells using the RNeasy™ Micro Kit Quick Start protocol following the manufacturer’s instructions. RNA concentration was quantified with a spectrophotometer NanoDrop© ND1000 (Thermo Fisher, Waltham, MA, USA). The quality of RNA was evaluated via the quotient absorption A_260_/A_280_. Further, RNA was reverse transcribed into cDNA following the RNeasy™ Micro Kit Quick Start protocol. Finally, quantitative PCR was performed using the Applied Biosystem. The primers used for each selected gene are listed in Table 1. The fold changes of infected HIBCPP cells with E-30 in relation to uninfected control, was calculated via the 2-ΔΔC_T_ methods. The results are represented as the fold change calculated via the 2-ΔΔC_T_ method using GAPDH as an internal control, and the relative fold change was determined between infected versus uninfected controls.

### 4.8. Massive Analysis of cDNA Ends (MACE) RNA Sequencing

We used MACE-seq and next-generation sequencing (NGS) to evaluate the transcriptomic changes in E-30 infected HIBCPP cells. Prior to RNA isolation, HIBCPP cells were rinsed twice in PBS. RNA was isolated from HIBCPP cells using the RNeasy™ Micro Kit Quick protocol according to the manufacturer’s instructions. We performed genome-wide gene expression profiling of all RNA samples (control inverted, standard control, inverted E-30 MOI 0.7, standard infected at MOI 0.7 and 20), using MACE-seq in order to identify potential transcriptomic changes upon E-30 infection. RNA sequencing libraries were analyzed using NGS. Fifteen MACE libraries were constructed using the MACE-Seq kit v. 2.0 (GenXPro GmbH, Frankfurt, Germany), according to the manufacturer’s protocol. MACE-Seq is a 3′-end targeted, tag-based, reduced representation transcriptome profiling technique, which can quantify all poly-adenylated transcripts. In general, the procedure follows a modified protocol described previously [74]. In brief, samples with 100 ng of DNase-treated RNA were used for library preparation. Synthesis of cDNA was performed by reverse transcription (RT) using oligo (dT) primers, following fragmentation of cDNA to an average size of 200 bp using sonification Bioruptor (Diagenode, Liège, Belgium). DNA was quantified using the Qubit HS dsDNA assay (Thermo Fisher Scientific, Waltham, MA, USA). Fragmented cDNA was ligated to DNA adapters containing *TrueQuant* unique molecular identifiers, included in the kit. Library amplification was done using polymerase chain reaction (PCR), purified by solid-phase reversible immobilization beads (Agencourt AMPure XP, Beckman Coulter, Carlsbad CA, USA) and subsequent sequencing was performed using a NextSeq platform (Illumina Inc., San Diego, CA, USA).

### 4.9. Statistical Analysis

Statistical analysis of functional and morphological polar infection of E-30 experiments was conducted using the Student’s *t*-test with GraphPad Quickcalcs online software (GraphPad Software, San Diego, CA, USA). Each one of the conditions, uninfected or infected with E-30 from low to high MOI, was considered as a fixed factor. Each mean per condition was compared two by two with the Student’s *t*-test. The figures are represented as mean ± SD.

Extensive bioinformatics analysis of MACE data was performed. Approximately 211 million MACE reads were obtained across all libraries. PCR-duplicates were identified using the TrueQuant technology and subsequently removed from raw data. Remaining reads were further poly (A)-trimmed and low-quality reads were discarded. In the following step, clean reads were aligned to the human reference genome (hg38, http://genome.ucsc.edu/cgi-bin/hgTables) using the bowtie2 mapping tool [75]. This alignment resulted in a gene dataset of a total of 32,272 different genes, for which the gene count data were normalized to account for differences in sequencing depth. In order to identify differential gene expression, statistical analysis was conducted using the statistical programming platform R (www.rproject.org/), using the DeSeq2 R/Bioconductor package [76]. As a result, *p*-value and log2-fold change (log2FC), were obtained for each gene in every comparison. False discovery rate (FDR) analysis was estimated to account for multiple testing. Genes with *p* < 0.05 and │log2FC│> 1 were considered to be differentially expressed. KOBAS (KEGG Orthology Based Annotation System, v. 3.0) was used to perform gene set enrichment analysis (GSEA), to identify over-represented biochemical pathways from 4 databases (KEGG PATHWAY, Reactome, PANTHER and Gene Ontology), and to calculate the statistical significance of each pathway.

### 4.10. Availability of Data 

The transcriptomic datasets generated and analyzed during the current study are available in the Gene Expression Omnibus (GEO) repository [77] and are accessible through GEO accession number GSE146890. The login password is: mpyfcsgqbzqdlar.

## 5. Conclusions

Polar infection of E-30 caused differential barrier affection and gene regulation in HIBCPP cells. The basolateral side was shown to be more susceptible to barrier alteration, infectivity and cytotoxicity, correlating with a potential in vivo situation, in which enterovirus access the CNS via the choroid plexus. Additionally, analysis of MACE RNA sequencing data of basolateral and apical infection compared to the uninfected condition revealed a significant upregulation of many genes implicated in host defense, such as *IFN*. Interestingly, some genes were differentially expressed depending on the side of infection, such as those associated with cytoskeleton remodeling and cell-cell adhesion. These findings highlight the importance of the HIBCPP cell side for effective E-30 infection. Future studies analyzing the signaling pathways involving the up- and downregulated genes will provide new insights into cellular processes that are differentially impacted in either basolaterally or apically infected HIBCPP cells.

## Figures and Tables

**Figure 1 ijms-21-06268-f001:**
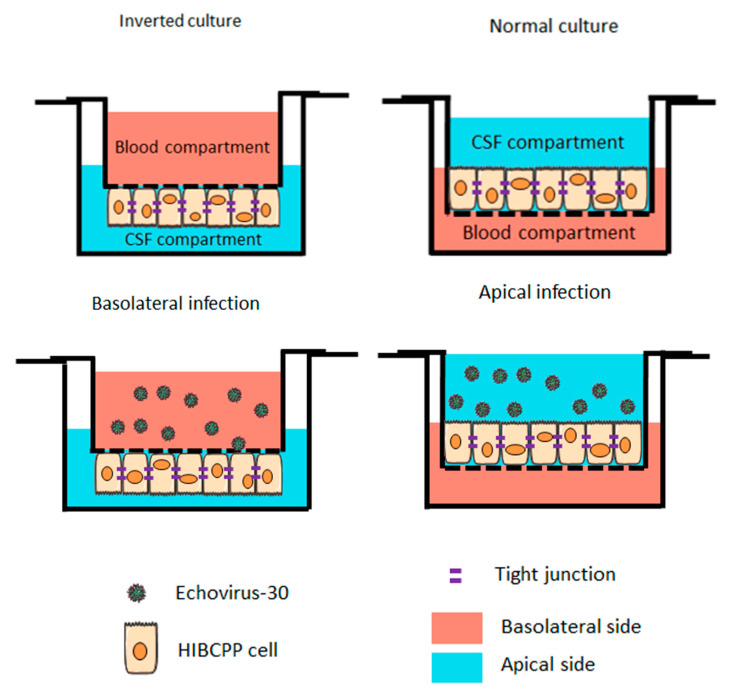
Schematic representation of infections with Echovirus 30 on the basolateral and apical side of the human choroid plexus papilloma (HIBCPP) cells. Schematic representation of HIBCPP cells in inverted culture (**left**) versus a normal culture (**right**). The HIBCPP cells are infected with E-30 on their apical side in normal culture, whereas they are infected on their basolateral side in inverted cultures.

**Figure 2 ijms-21-06268-f002:**
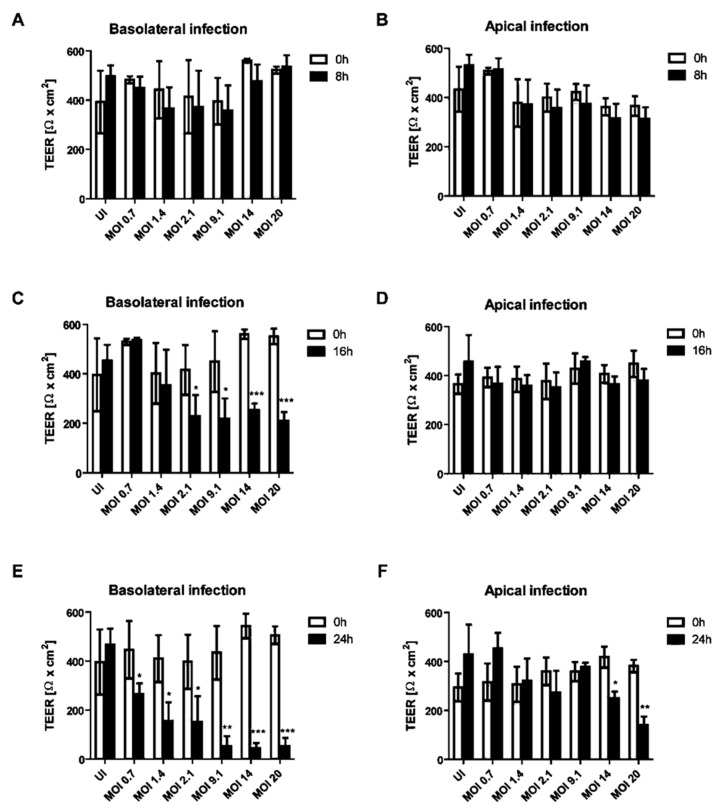
Decrease of transepithelial electrical resistance (TEER) of HIBCPP cells is dependent on the side of infection with E-30. Barrier integrity of HIBCPP cells layer was evaluated following basolateral (T = 8 h (**A**), T = 16 h (**C**), T = 24 h (**E**)) and apical (T = 8 h (**B**), T = 16 h (**D**), T = 24 h (**F**)) infection with E-30 via measurements of the transepithelial electrical resistance (TEER), for all investigated conditions (uninfected (UI) and infected with MOI 0.7, MOI 1.4, MOI 2.1, MOI 9.1, MOI 14 and MOI 20). Data are shown as mean ± SD of 3 independent experiments, each performed in triplicates. Statistical comparisons with *p*-values are shown for T = 8 h, T = 16 h and T = 24 h compared to the respective TEER at T = 0 h for each MOI or the uninfected control (=UI). For statistical analysis, the Student’s *t*-test was used. *p*-values are displayed as follows: * *p* < 0.05; ** *p* < 0.001 and *** *p* < 0.0001. The statistical analysis was performed with the software GraphPad QuickCalcs.

**Figure 3 ijms-21-06268-f003:**
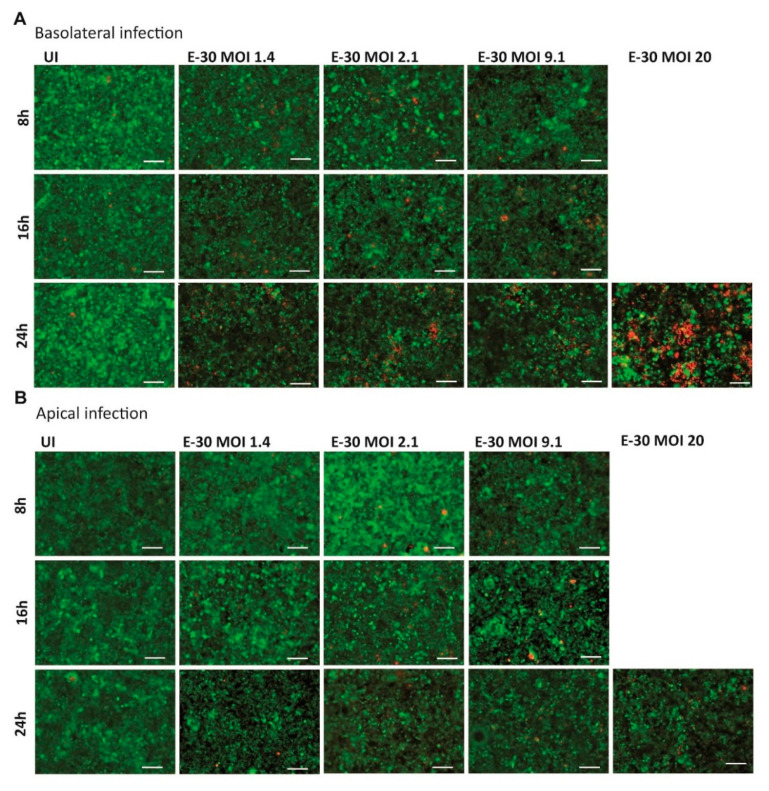
E-30 infection from the basolateral side of HIBCPP cells has a greater cytotoxic effect. The Live/Dead viability assay stained the living cells in green (cell tracker green) and the dead cells in red (ethidium homodimer-1). Images of HIBCPP cells infected from the basolateral side (**A**) and the apical side (**B**), for 8, 16 and 24 h and for each condition (uninfected (UI), and with E-30 infection at MOI 1.4, 2.1, 9.1 and 20). The pictures show representative images from three experiments, each performed in duplicates. White scale bars represent 100 µM.

**Figure 4 ijms-21-06268-f004:**
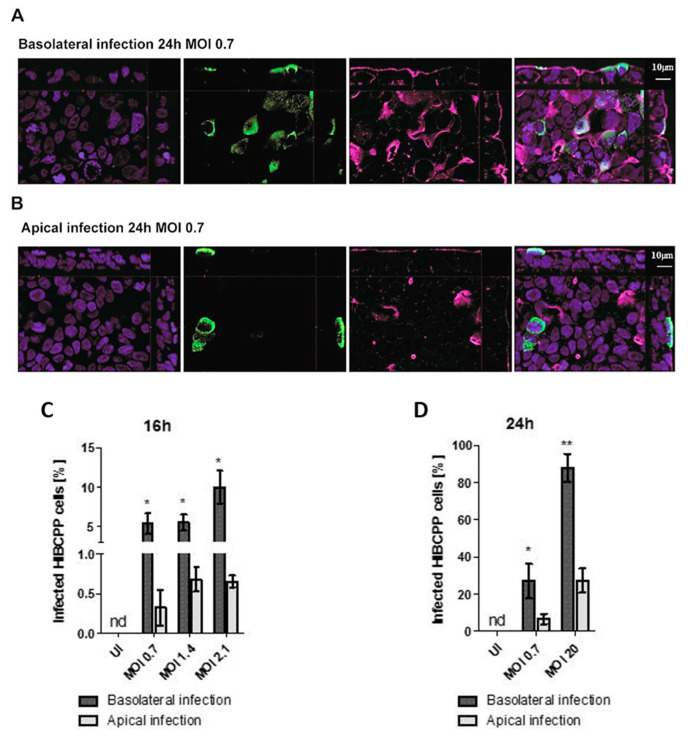
The number of infected HIBCPP cells following basolateral infection is greater compared to apical infection with E-30 at MOI 0.7 for 24 h. Immunofluorescence imaging of HIBCPP cell layers following 24 h of infection with E-30 on the basolateral side (**A**) or apical side (**B**). Nuclei are stained with DAPI (blue), HIBCPP cells infected with E-30 are stained with anti-PAN entero (green), and actin is stained with phalloidin (purple). Stacks were acquired using Zeiss Apotome and Zen software with an X63/1.4 objective lens. Scale bars are indicated in the figure. Quantification of HIBCPP cells with E-30 infection from the basolateral and apical side following 16 h (**C**) and 24 h (**D**) with different MOI (MOI 0.7, 1.4, 2.1 and MOI 0.7, 20, respectively). The data are shown as mean ± SD of 3 independent experiments, each performed in duplicates. Statistical significance was calculated using a Student’s *t*-test. *p*-values are displayed as follows; * *p* < 0.05 and ** *p* < 0.001. The statistical analysis was performed with the software GraphPad QuickCalcs.

**Figure 5 ijms-21-06268-f005:**
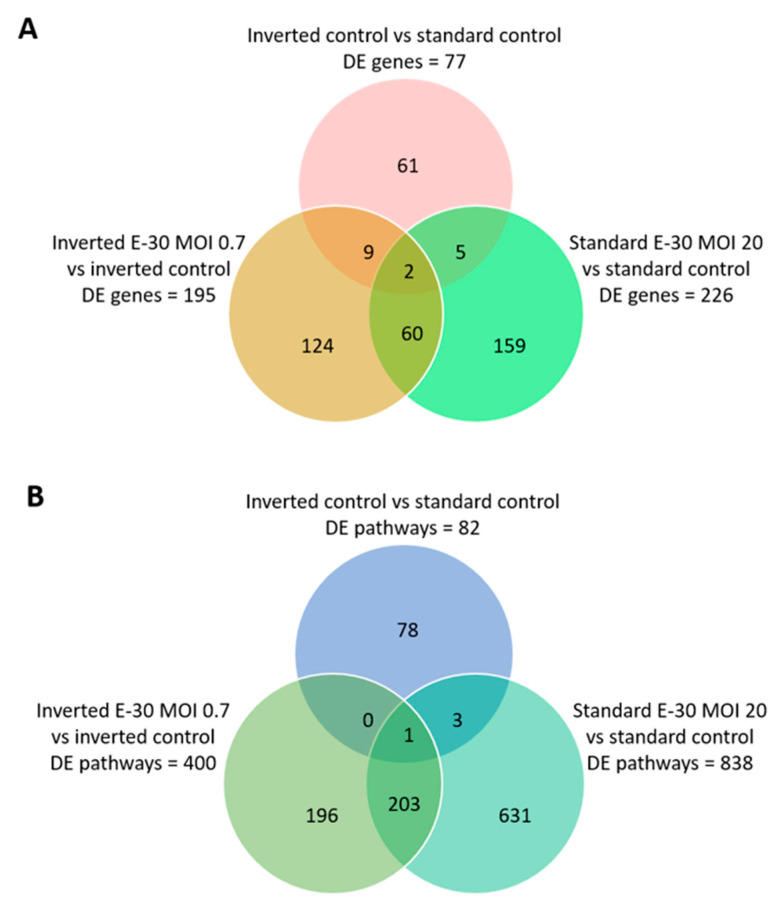
Differentially expressed genes and pathways following polar E-30 infection of HIBCPP cells. Venn diagram representing differentially expressed genes (**A**) and pathways (**B**) following infection of HIBCPP cells with E-30 from the apical (standard E-30 MOI 20) or basolateral side (inverted E-30 MOI 0.7), versus the respective uninfected control conditions (inverted control and standard control). Comparisons resulted in a *p*-value and log2-fold change (log2FC) for every gene for the 3 experiments. Genes with *p* < 0.05, and a │log2FC│ > 1 were considered as differentially expressed (*n* = 3).

**Figure 6 ijms-21-06268-f006:**
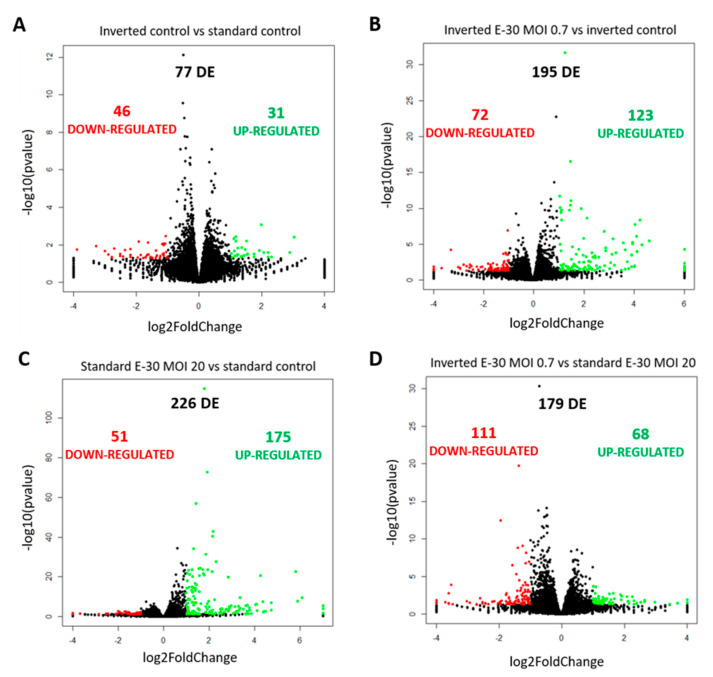
Polar E-30 infection of HIBCPP cells has a differential impact on HIBCPP cells’ RNA. Volcano plots representing genes found upregulated (in green) and downregulated (in red) in the inverted control versus the standard control (**A**), the inverted culture infected with E-30 MOI 0.7 versus the inverted control (**B**), the standard culture infected with E-30 MOI 20 versus the standard control (**C**) and the inverted culture infected with E-30 MOI 0.7 versus the standard culture infected with E-30 MOI 20 (**D**). Comparisons resulted in a *p*-value and log2FC for every gene for the 3 experiments. Genes with *p* < 0.05, and a │log2FC│ > 1 were considered as differentially expressed. Statistical analysis was performed using the R programming platform using the DeSeq2 R/Bioconductor package.

**Figure 7 ijms-21-06268-f007:**
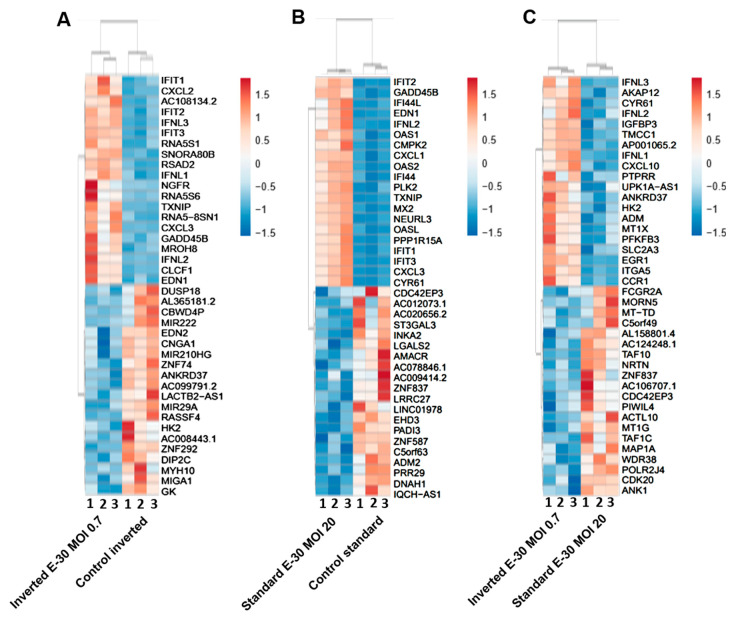
Polar E-30 infection of HIBCPP cells has a differential impact on HIBCPP cells’ RNA. Heatmaps representing the first 20 upregulated (in red) and 20 downregulated genes (in blue), in inverted culture infected with E-30 MOI 0.7 versus inverted control (**A**), standard culture infected with E-30 MOI 20 versus standard control (**B**), inverted culture infected with E-30 MOI 0.7 versus standard culture infected with E-30 MOI 20 (**C**). This resulted in a *p*-value and log2-fold change (log2FC) for every gene for the 3 experiments. Genes with *p* < 0.05, and a │log2FC│ > 1 were considered as differentially expressed. Statistical analysis was performed using the R programming platform using the DeSeq2 R/Bioconductor package.

**Figure 8 ijms-21-06268-f008:**
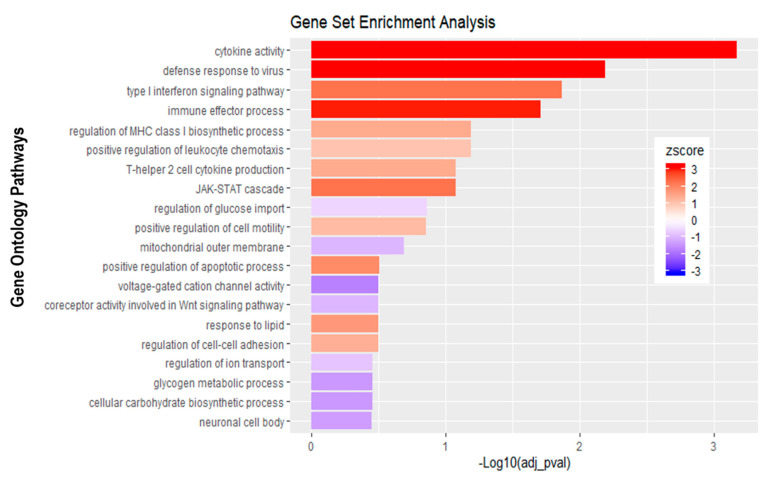
Gene set enrichment analysis of basolateral infection with E-30 MOI 0.7 versus inverted control. Bar chart representing the top 20 significantly enriched pathways in inverted E-30 MOI 0.7 infection compared to inverted control. Analysis was performed using KOBAS 3.0. This resulted in Benjamini–Hochberg corrected *p*-value and z-score (calculated as suggested by [26]).

**Figure 9 ijms-21-06268-f009:**
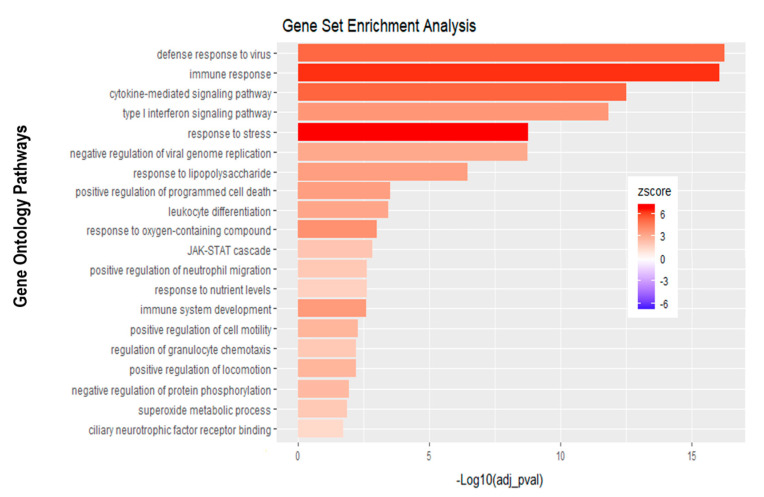
Gene set enrichment analysis of apical infection with E-30 MOI 20 versus control condition reveals differentially expressed pathways. The bar chart represents the top 20 significantly enriched pathways in apical E-30 MOI 20 infection compared to standard control. Analysis was performed using KOBAS, 3.0. This resulted in Benjamini–Hochberg corrected *p*-value and z-score calculated as suggested by [26].

**Figure 10 ijms-21-06268-f010:**
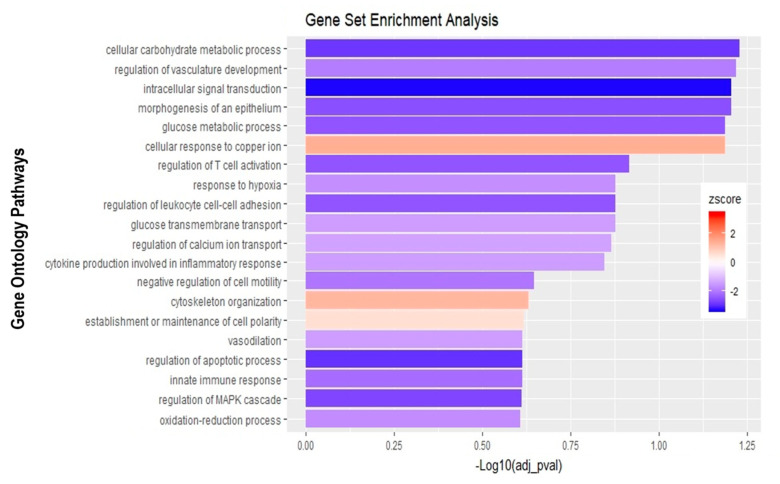
Gene set enrichment analysis of basolateral infection with E-30 MOI 0,7 versus apical infection with E-30 MOI 20. Bar chart representing the top 20 significantly enriched pathways in basolateral E-30 MOI 0,7 compared to apical E-30 MOI 20 infection. Analysis was performed using kobas 3.0. This resulted in Benjamini–Hochberg corrected *p*-value and z-score calculated as suggested by [26].

**Figure 11 ijms-21-06268-f011:**
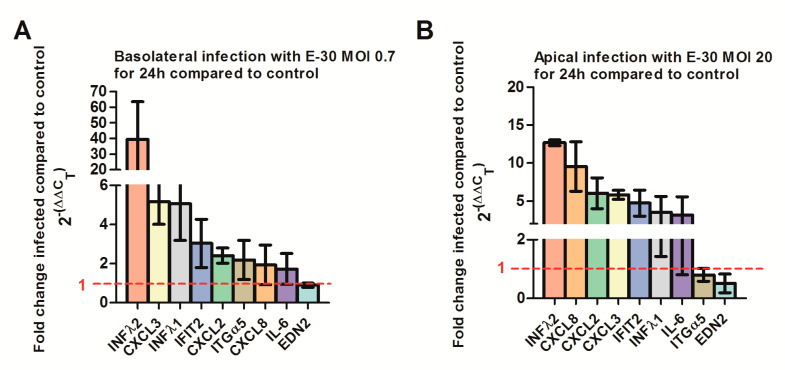
Quantitative PCR determining the regulation of CXCL2, 3, 8, IL-6, IFNλ1, IFNλ2, ITGα5, EDN2 and IFIT2 genes following apical or basolateral infection of HIBCPP cells with E-30. The histograms represent the fold change of HIBCPP cells infected with E-30 in relation to the uninfected control. The fold change was calculated via the 2-ΔΔC_T_ method using GAPDH as an internal control, and the relative fold change was determined between infected versus uninfected controls. (**A**) Fold change of HIBCPP cell genes following basolateral infection with E-30 at MOI 0.7 for 24 h, (**B**) fold change of HIBCPP cell genes following apical infection with E-30 at MOI 20 for 24 h. Data are shown as mean ± SD of 3 independent experiments, each performed in triplicates.

**Table 1 ijms-21-06268-t001:** Primers used for RT-PCR.

Genes	Primer Forward	Primer Reverse
CXCL3	5′-CGCCCAAACCGAAGTCATAG-′3	3′-GCTCCCCTTGTTCAGTATCTTTT-′5
CXCL2	5′-CTCAAGAATGGGCAGAAAGC-′3	3′-AAACACATTAGGCGCAATCC-′5
IL-6	5′-AACCTGAACCTTCCAAAGATGG-′3	3′-GTCAGGGGTGGTTATTGCAT-′5
GAPDH	5′-TGTTGCCATCAATGACCCCTT-′3	3′-CTCCACGACGTACTCAGCG-′5
IFNλ1	5′-TGTCACCTTCAACCTCTTCCG-′3	3′-TAAGGTGTGGGGTGTCAGGT-′5
IFNλ2	5′-ACATCCCAGACAGAGCTCAAAA-′3	3′-CCAGGGTCTGTTTGGGTCTT-′5
IFIT2	5′-AGCGAAGGTGTGCTTTGAGA-′3	3′-AGGGTCAATGGCGTTCTGAG-′5
EDN2	5′-CGTCCTCATCTCATGCCCAA-′3	3′-GCCGTAAGGAGCTGTCTGTT-′5
ITGα5	5′-CGCCTCTGGGAGGTTTAGGA-′3	3′-TCTAAGTTGAAGCCCCCGAC-′5
CXCL8	5′-AGGAAAACTGGGTGCAGAGG-′3	3′-TGCTTGAAGTTTCACTGGCATC-′5

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
