# Peer review of "Polar Infection of Echovirus-30 Causes Differential Barrier Affection and Gene Regulation at the Blood–Cerebrospinal Fluid Barrier"

_ijms, 2020, doi:10.3390/ijms21176268_

Round 1

Reviewer 1 Report

Wiatr et al. investigate the infection of polar cells (such as cells of the choroid plexus) with a relevant human pathogen, the Echovirus-30, which can cause severe to fatal infections of the brain, esp. in children. The last months have clearly shown the importance of infectiology research.

The blood-brain barrier (BBB) and the blood-cerebrospinal fluid barrier (bCSFb) provide important control and protection barriers for access of molecules and harmful into the brain and CSF, which comprise sterile and immune compromised areas.

In this study, HIBCPP cells were used as a model for the bCSFb and differing effects (epithelial integrity, cytotoxicity, infection and gene expression) depending on the location of virus access were analysed. Although this is an established model for the investigation of the bCSFb, and the cells possess properties of epithelial tissue (polarity, tight junctions, transepithelial resistance), the authors should handle the results derived from this model with care. Immortalized cells are prone to undergo functional changes due to intense passaging, and lack of the basement membrane, which is present in the bCSFb, could also lead to a blurred picture of E-30 infection properties. The authors should also point out the limitations of their model.

Although in my opinion the study is generally well designed, and presentation of results and English language are fine, there are a few points that should be addressed by the authors and details should be mentioned or corrected in the corresponding sections.

  • My major concern is the uncertainty about the patho-physiological circumstances the virus particles enter the cell. The authors hypothesize that infection from basolateral as well as from apical side is possible. Although this seems not to be proven, it is the key idea of this study. Therefore, the authors should explain in more detail what is known from literature in the introduction section, provide information about the structure of the barrier and discuss results received from the in vitro model more carefully and more moderately.

Smaller issues/questions that in my opinion should be addressed/corrected by the authors:

  • Does E-30 only use the bCSFb route or also the BBB? How would infection from the apical side be possible?
  • Can the authors mention why they decided for the used MOIs?
  • Could the authors provide an evaluation (graph) of the imaging of the cytotoxicity assay (Fig 3)?
  • Methods section: the virus preparation should be shortly described
  • Are there other models of the bCSFb available? (for discussion, not for further experiments)

Author Response

Reviewer 1:

Wiatr et al. investigate the infection of polar cells (such as cells of the choroid plexus) with a relevant human pathogen, the Echovirus-30, which can cause severe to fatal infections of the brain, esp. in children. The last months have clearly shown the importance of infectiology research.

The blood-brain barrier (BBB) and the blood-cerebrospinal fluid barrier (bCSFb) provide important control and protection barriers for access of molecules and harmful into the brain and CSF, which comprise sterile and immune compromised areas.

In this study, HIBCPP cells were used as a model for the bCSFb and differing effects (epithelial integrity, cytotoxicity, infection and gene expression) depending on the location of virus access were analysed. Although this is an established model for the investigation of the bCSFb, and the cells possess properties of epithelial tissue (polarity, tight junctions, transepithelial resistance), the authors should handle the results derived from this model with care. Immortalized cells are prone to undergo functional changes due to intense passaging, and lack of the basement membrane, which is present in the bCSFb, could also lead to a blurred picture of E-30 infection properties. The authors should also point out the limitations of their model.

Response: We thank the reviewer for this comment. We therefore added an additional part in the discussion section of the mansucript. Moreover, other BCSFB models were mentioned in more detail as also requested further below by the same reviewer.

Page 14-line 312-315„The limitation of our model is that we used HIBCPP cells as a BCSFB model and not primary human choroid plexus epithelial cells. However, our model is currently the only human model existing. Still, HIBCPP may respresent not all properties as primary cells and therefore our results should be interpreted with caution.“ 

Although in my opinion the study is generally well designed, and presentation of results and English language are fine, there are a few points that should be addressed by the authors and details should be mentioned or corrected in the corresponding sections.

  • My major concern is the uncertainty about the patho-physiological circumstances the virus particles enter the cell. The authors hypothesize that infection from basolateral as well as from apical side is possible. Although this seems not to be proven, it is the key idea of this study. Therefore, the authors should explain in more detail what is known from literature in the introduction section, provide information about the structure of the barrier and discuss results received from the in vitro model more carefully and more moderately.

Response: In the publication Schneider et a. 2012 (Virus Research) could already proof apical and basolateral infection of HIBCPP with E-30. Therefore, we did not repeat these published experiments in the current manuscript. Hence, we underlinded this point in the manuscript, now and have added the other abovementioned results into the introduction section of the manuscript (Page 2, line 64-65). We further added the aspect of limitations a recommended above.

Schneider H, Weber CE, Schoeller J, Steinmann U, Borkowski J, Ishikawa H, Findeisen P, Adams O, Doerries R, Schwerk C, Schroten H, Tenenbaum T. Chemotaxis of T-cells after infection of human choroid plexus papilloma cells with Echovirus 30 in an in vitro model of the blood-cerebrospinal fluid barrier. Virus Res. 2012 Dec;170(1-2):66-74.

Smaller issues/questions that in my opinion should be addressed/corrected by the authors:

  • Does E-30 only use the bCSFb route or also the BBB? How would infection from the apical side be possible?

Response: „In human BBB models applying human brain microvascular endothelial cells (HBMEC) or human cerebral microvascular endothelial cells (hCMEC/D3), it was shown that infection with poliovirus [Coyne et al., 2007], E-6, E-12 and E-30 [Volle et al., 2015] resulted in damage to the junctional connections, leading to increased paracellular permeability of the barrier. Therefore, enterovirus that has crossed the BBB and is cirulating in the CSF may apically enter the choroid plexus epithelium. Moreover, a reseeding of enterovirus that have entered via the BSCFB into the bloodstream is also feasible.

This information was added to the discussion section of the manuscript (Page 14, line 296-302).

Coyne CB, Kim KS, Bergelson JM: Poliovirus entry into human brain microvascular cells requires receptor-induced activation of SHP-2. EMBO J 2007, 26:4016-4028.

Volle R, Archimbaud C, Couraud PO, Romero IA, Weksler B, Mirand A, Pereira B, Henquell C, Peigue-Lafeuille H, Bailly JL: Differential permissivity of human cerebrovascular endothelial cells to enterovirus infection and specificities of serotype EV-A71 in crossing an in vitro model of thehuman blood-brain barrier. J Gen Virol 2015, 96:1682-1695.

  • Can the authors mention why they decided for the used MOIs?

Response: The use of the specific MOIs was the result of an experimental decision to find an optimal infection dose. The efficacy of the use of the MOI 0.7 was also previously published. Lastly, the MOI number is the result of the quantitive analysis of the enterovirus qPCR.

  • Could the authors provide an evaluation (graph) of the imaging of the cytotoxicity assay (Fig 3)?

Response: Unfortunatly, the images were only optically analysed. Since all images show only minor and comparable cytotoxicity in all experimental conditions, there is in our opinion no real need for a quantitative analysis. The only obvious major cytotoxicity occured with a MOI 20 after the basolateral infection. This was already clearly stated in the manuscript and not used for further experiments.

  • Methods section: the virus preparation should be shortly described

Response: The virus preparation has extensively been described on page 17 line 431-440 (in the original publication as well (Schneider et al., 2012). Therefore, we did not further add new aspects.

  • Are there other models of the bCSFb available? (for discussion, not for further experiments)

Response: There is currently no other human BCSFB barrier model available. Several other models have been described.

The following text was included (added on page 14, line 315-320):

Several rodent models of BCSFB have been established, including immortalized rat, as well as primary mouse choroid plexus epithelial cells (Monnot and Zheng, 2013; Castro Dias et al., 2019). Primary porcine choroid plexus epithelial cell cultures have been used to study bacterial infection of the brain and have demonstrated polar infection (Tenenbaum et al., 2009). A novel porcine choroid plexus epithelial cell line (PCP-R) has also been established (Schroten et al., 2012). Moreover, BCSFB model with epithelial cells from Indian origin Rhesus macaques has recently been described (Delery and Macean, 2019).

Monnot, A. D., and Zheng, W. (2013). Culture of choroid plexus epithelial cells and in vitro model of blood–CSF barrier. Methods Mol. Biol. 945, 13–29.

Castro Dias M, Coisne C, Lazarevic I, Baden P, Hata M, Iwamoto N, Francisco DMF, Vanlandewijck M, He L, Baier FA, Stroka D, Bruggmann R, Lyck R, Enzmann G, Deutsch U, Betsholtz C, Furuse M, Tsukita S, Engelhardt B. Claudin-3-deficient C57BL/6J mice display intact brain barriers. Sci Rep. 2019 Jan 18;9(1):203.

Tenenbaum T, Papandreou T, Gellrich D, Friedrichs U, Seibt A, Adam R, Wewer C, Galla HJ, Schwerk C, Schroten H. Polar bacterial invasion and translocation of Streptococcus suis across the blood-cerebrospinal fluid barrier in vitro. Cell Microbiol. 2009 Feb;11(2):323-36.

Schroten M, Hanisch FG, Quednau N, Stump C, Riebe R, Lenk M, Wolburg H, Tenenbaum T, Schwerk C. A novel porcine in vitro model of the blood-cerebrospinal fluid barrier with strong barrier function. PLoS One. 2012;7(6):e39835

Delery EC, MacLean AG. Culture Model for Non-human Primate Choroid Plexus. Front Cell Neurosci. 2019 Aug 28;13:396.

Reviewer 2 Report

The study investigated polar infection of human choroid plexus papilloma (HIBCPP) cells with Echovirus-30 (E-30). However, there are few minor comments required to be corrected:

Page 1; Line 30 in Abstract section: " significantly higher compared " this results should be given in figures or statistical significance values.

Page 2; Line 62: "manuscript" should be replaced with "study"

Page 16; Line 409: The chemicals and kits were used in the study should be given in a separate subsection.

Page 19; Line 512: The subheading should be changed from "Statistics" to "Statistical analysis"

Author Response

Reviewer 2:

The study investigated polar infection of human choroid plexus papilloma (HIBCPP) cells with Echovirus-30 (E-30). However, there are few minor comments required to be corrected:

Page 1; Line 30 in Abstract section: " significantly higher compared " this results should be given in figures or statistical significance values.

Response: The requested infomation could not be included in the abstract, because we have already have reached the word limitation of 200 words.

Page 2; Line 62: "manuscript" should be replaced with "study"

Response: The modification was made according to the reviewer`s recommendation.

Page 16; Line 409: The chemicals and kits were used in the study should be given in a separate subsection.

Response: An additional section was created as recommended by the reviewer.

Page 19; Line 512: The subheading should be changed from "Statistics" to "Statistical analysis"

Response: The modification was made according to the reviewer`s recommendation.